# A Retrospective Analysis of White Clover (*Trifolium repens* L.) Content Fluctuation in Perennial Ryegrass (*Lolium perenne* L.) Swards under 4 Years of Intensive Rotational Dairy Grazing

Áine Murray [1,2], Luc Delaby [3], Trevor J. Gilliland [2], Bríd McClearn [1], Michael Dineen [1], Clare Guy [1] and Brian McCarthy [1,*]

1 Teagasc, Animal & Grassland Research and Innovation Centre, Moorepark, P61 C996 Fermoy, Ireland; aine.murray@teagasc.ie (Á.M.); brid.mcclearn@teagasc.ie (B.M.); michael.dineen@teagasc.ie (M.D.); stephclareguy@gmail.com (C.G.)

2 The Institute of Global Food Security, Queen's University Belfast, BT7 1NN Belfast, Ireland; tj.gill@hotmail.co.uk

3 INRAE, Institut Agro, Physiologie, Environnement et Génétique Pour l'Animal et les Systèmes d'Elevage, F-35590 Saint-Gilles, France; luc.delaby@inrae.fr

* Correspondence: brian.mccarthy@teagasc.ie

**Abstract:** The objective of this study was to examine fluctuations in white clover (*Trifolium repens* L.) content in perennial ryegrass (*Lolium perenne* L.) swards within a high nitrogen (250 kg N/ha) input grazing dairy system. The data came from a larger, overall system experiment within which all management and growing condition variables were categorised each year for the 40 paddocks that contained perennial ryegrass-white clover swards, over four growing years. Within that study, eight perennial ryegrass cultivars were examined, each sown individually with two white clover cultivars in a 50:50 mix of 'Chieftain' and 'Crusader'. To determine management associations and meteorological patterns with white clover content and rate/direction of change, separate generalised linear models were used to analyse each individual management or meteorological variable. Paddocks with high white clover contents were associated with lower pre- and post-grazing sward heights, lower pasture cover over the winter period and shorter over-winter period. Perennial ryegrass cultivars with lower pre- and post-grazing height, lower pre-grazing pasture mass and pasture yield removed, all retained more white clover in their swards. Soil fertility remained a key factor that affected white clover persistence influencing the degree of responses in all treatments, particularly soil phosphorus (*P*) levels. Beyond this, higher white clover contents and lower rates of white clover decline were associated with paddocks that received lower rainfall, had higher soil moisture deficits and received more radiation into the base of the sward, particularly around the time of grazing.

**Keywords:** white clover; perennial ryegrass; persistency; cultivar; cattle grazing

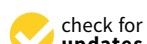

## 1. Introduction

Perennial ryegrass (*Lolium perenne* L.; PRG) pasture-based ruminant systems are highly efficient with low input cost but are ultimately dependent on high inorganic nitrogen (N) levels to support pasture growth [1,2]. To reduce production costs and environmental impacts of inorganic N use [3] and to increase farm gate N use efficiency (NUE—[4]), there is renewed interest in the incorporation of legumes, in particular white clover (*Trifolium repens* L.; WC), in PRG pasture-based production systems [5]. White clover possesses the ability to biologically fix between 10–300 kg N/ha from the atmosphere, proportional to the amount of legume grown [6–8], into a plant usable form (i.e., nitrate) to facilitate pasture growth [9] while allowing a reduction in inorganic N fertiliser use [10]. White clover is a nutritionally superior forage to PRG [11,12] as it promotes higher dry matter intake due to its lower neutral detergent fibre (NDF) content, which can lead to higher

milk production per cow [13–15]. Consequently, WC is the predominant legume sown in temperate pastures [16] and is an excellent natural resource that can be incorporated into PRG swards to increase animal production [11,17,18] and efficiency of pasture-based ruminant production.

It has been reported that WC needs to be present within the sward at a minimum proportion of 20% in order to see production benefits [19], with animals selecting diets containing up to 50–60% WC within the sward [20]. However, sward WC content and persistency are often erratic both within and across years and between paddocks within a farm [17,21]. Previous research has shown that sward WC content fluctuates over time and that a number of key soil, climatic and management factors are associated with these fluctuations [22–24]. Soil factors, such as pH, are important in ensuring successful establishment, as WC has a preference for slightly acidic or neutral pH values [16]. Issues around WC competitiveness in PRG-WC mixtures have been strongly linked to management, with one of the major factors being soil N [23]. Increased inorganic N fertiliser has been shown to have a negative effect on the prevalence and productivity of WC in the sward through increased competition from the companion grass [17,25,26]. This has raised doubts around how much benefit WC brings to intensively managed pastures receiving higher levels of N fertilizer.

If WC is to have a key role in pastures that support high stocking rates and high animal performance, a greater understanding of the dynamics between soil, management and meteorological factors under such intensive management is still required. The underlying challenge is how to optimise WC content in ruminant production systems that require high pasture productivity, generally driven by high N fertilizer use, and whether this can be achieved as a sustainable practice. Recently, WC was incorporated with PRG under an intensive animal grazing system with high N fertilizer levels [18]. Over the course of four years, the WC content of the sward diminished from an average of 31.7% in year one to 15.4% in year four. The objective of the current study was to undertake a retrospective analysis of the soil, management and meteorological factors in the study of McClearn et al. [18] to ascertain if these were associated with WC persistency. It was hypothesised that high WC contents and persistency could be maintained under an intensive cattle-grazing regime when soil, management and meteorological factors were in favour of WC, to help it compete with PRG.

## 2. Materials and Methods

The data for the current analysis came from a larger overall system experiment that examined the effect of PRG tetraploid and diploid cultivars sown individually with and without WC and managed at the same stocking rate and N fertilization level, on the productivity of a spring-calving pasture-based dairy system [18]. This experiment was undertaken from 2014 to 2017 in Teagasc Clonakilty Agricultural College, Co. Cork, Ireland (51°63′ N, 08°85′ E; 25–70 m above sea level). The current analysis only utilised data from the PRG-WC swards examined by McClearn et al. [18] and thus comprised two separate farmlets, one based on tetraploid-WC and the other on diploid-WC paddocks. The first farmlet comprised 4 tetraploid cultivars (Astonenergy, Dunluce, Kintyre, and Twymax, sown at 37.5 kg/ha) each in separate paddocks with two medium-leaved WC cultivars (a 50:50 mix of Chieftain and Crusader, sown at 5 kg/ha). This was repeated 5 times to give a total of 20 paddocks in the farmlet. The other farmlet comprised the same configuration but using 4 diploid cultivars (AberChoice, Glenveagh, Tyrella, and Drumbo, sown at 30 kg/ha). The experimental area was sown in 2012 (75%) and 2013 (25%) by full cultivation (ploughing and tilling). Thirty spring-calving dairy cows were assigned to each 10.9 ha farmlet every year, giving a stocking rate of 2.75 cows/ha (paddocks ranged in size from 0.43 to 0.71/ha).

### 2.1. Grazing Management

The farmlets were managed in a spring-calving rotational grazing system whereby cows grazed both day and night as they calved from February (weather permitting).

Typically, the grazing season began in early February and finished in mid-November each year. Cows were supplemented with 4 kg of concentrate post-calving, and this supplement was gradually reduced as pasture growth on the treatments met herd feed requirements, typically in mid-April each year.

Weekly monitoring of average farm cover was undertaken for each individual treatment using the online application PastureBase Ireland (PBI; [27]), which was the main decision support tool used to manage grazing. Target pre-grazing pasture mass (PrGPM) was calculated separately for each grazing treatment using the formula as found in the Teagasc Dairy Manual: Target PrGPM = (stocking rate (on grazing platform) × ideal rotation length × daily pasture allowance per cow) + residual pasture mass, where ideal rotation length during main grazing season (April to July) = 21 d and daily pasture allowance (>4.0 cm) = 17 to 18 kg DM/cow per d [28].

Residency time within paddocks was dictated by a target post-grazing sward height (PoGSH) of 3.5 to 4.0 cm for the first and last grazing rotation and 4.0 cm during the main grazing season. When the required PoGSH was reached, cows within treatments were moved to their subsequent paddocks. No mechanical correction, by mowing of paddocks post-grazing, took place during the 4-year study, and all excess forage was cut and conserved as silage. Silage yields were measured by quadrant cuts and sward WC content was not measured from paddocks when silage was harvested. Inorganic N was applied equally across both sward types in the form of urea or calcium ammonium nitrate at a rate of 250 kg of N/ha per year. Swards received 28 kg N/ha in late January, 30 kg N/ha in mid-March, 30 kg N/ha after the second and third rotation, and 20 kg N/ha for every rotation in succession until the period for fertilizer spreading closed on the September 15 each year. If paddocks were selected to be closed for first cut silage, they received an elevated level of N of 112 kg N/ha after the first grazing. Inorganic phosphorus (P) and potassium (K) were applied across all swards based on yearly soil test results.

### 2.2. Pasture Measurements

Grazing data were gathered at each grazing event for each individual treatment. Pre-grazing pasture mass was measured prior to grazing by harvesting 2 strips (approximately 10 m × 1.2 m) to a post-height of 4.0 cm using an Etesia mower (Etesia UK Ltd., Warwick, UK). The harvested forage was weighed and a 100 g subsample was dried at 90 °C for 15 h to determine DM. Ten sward heights were taken before and after each strip of forage was harvested, using a rising platemeter (Jenquip, Feilding, New Zealand). This was used to calculate sward density by the following equation: Sward density (kg of DM/cm/ha) = PrGPM/ (precutting height − postcutting height).

Pre-grazing sward height (PrGSH) and PoGSH were calculated across whole paddocks before and after grazing using a platemeter taking compressed sward heights at 30 locations pre-grazing and 50 locations following grazing. Pre-grazing pasture mass above 4 cm was calculated using sward density according to the following equation: Pre-grazing pasture mass above 4 cm (kg DM/ha) = (PrGSH − 4 cm) × sward density [29]. Pasture removed and grazing efficiency were calculated as follows: Pasture removed (kg DM/ha) = (PrGSH − PoGSH) × sward density; Grazing efficiency (%) = (pasture removed /PrGPM) × 100.

Pasture production was categorised as either grazing pasture production or silage production and was summed by paddock to give total pasture production [27]. All pasture production was recorded and calculated using the online application PBI.

### 2.3. White Clover Content

White clover content was estimated in each paddock before every grazing event. A Gardena hand shears (Accu 60; Gardena International GmbH, Ulm, Germany) was used to take 15 evenly distributed pasture snip samples cut to 4 cm across the paddock. The fresh pasture sample was mixed, and two 70 g cut-samples were weighed and separated by

hand into WC, PRG, and other plant material. The samples were dried at 60 °C for 48 h to determine proportions on a DM basis.

### 2.4. Meteorological Data

Meteorological data were collected by a weather station 7 km away at Timoleague, (Agricultural Catchments Program, Teagasc). Data recorded comprised of daily rainfall (mm), mean, maximum and minimum air temperature (°C), wind speed (m/s), solar radiation (J) and soil temperature (°C) and are presented in Table 1. From these data, soil moisture deficit (SMD; mm) was calculated $SMD_t = SMD_{t-1} - Rain_t + ET_t + Drain$ where $SMD_t$ and $SMD_{t1}$ are the SMD on day t and day t-1, respectively (mm), Rain is the daily precipitation (mm/day), $ET_t$ the daily actual evapotranspiration (mm/day), and Drain equals the amount of water drained daily (mm/day) by percolation and/or overland flow [30]. The data were divided into four time periods defined as; full year (1 January–31 December), early season (1 January–30 April), mid-season (1 May–31 August) and late season (1 September–31 December) and used to create the variables in the dataset. To investigate whether WC persistency was affected by meteorological conditions during the regrowth stage (the number of days between grazing events), the meteorological conditions during 7 days prior to and 7 days post-grazing, as well as during the grazing event were calculated and used as four additional variables

**Table 1.** Meteorological data for 2014–2017 compared with the 10-year average (2004–2014).

| Time Period | Full Year | Early Season | Mid-Season | Late Season |
|---|---|---|---|---|
| | | Air Temperature (°C) | | |
| 2014 | 10.5 | 8.1 | 14.8 | 10.4 |
| 2015 | 9.9 | 7.1 | 13.4 | 10.7 |
| 2016 | 10.0 | 7.5 | 14.5 | 9.7 |
| 2017 | 10.2 | 8.3 | 14.1 | 9.6 |
| 10-year average | 9.9 | 7.5 | 14.6 | 9.3 |
| | | Soil temperature (°C) | | |
| 2014 | 11.4 | 8.5 | 16.4 | 11.4 |
| 2015 | 11.3 | 8.4 | 15.5 | 11.8 |
| 2016 | 11.2 | 8.6 | 16.0 | 10.9 |
| 2017 | 11.6 | 9.3 | 15.9 | 11.2 |
| 10-year average | 10.8 | 8.1 | 16.6 | 9.8 |
| | | Rainfall (mm) | | |
| 2014 | 1094 | 448 | 181 | 465 |
| 2015 | 1468 | 434 | 301 | 733 |
| 2016 | 1012 | 524 | 224 | 264 |
| 2017 | 1143 | 367 | 292 | 484 |
| 10-year average | 1046 | 403 | 263 | 380 |

### 2.5. Paddock Management Factors

As described previously, data were recorded using PBI for each grazing/silage event of each paddock. The standard measurements recorded were as follows: date of grazing start, grazing end and residency time, date of silage event, PrGSH and PoGSH, PrGPM, pasture removed, DM content and sward density. Four different rotation length variables were created; the average rotation length (the average number of days between any harvesting (grazing or silage event); the over winter rotation length (the number of days between the last grazing in autumn to first grazing in spring); grazing rotation length (the average number of days between grazing events over the entire grazing season); silage rotation length (average number of days between a previous event—grazing or silage cut and a silage cut event). The average PrGPM and average pasture removed from a paddock at each grazing was calculated for the four time periods. The performance of each individual paddock was also calculated for the total number of grazing days per ha and the total pasture removed per ha by grazing, as silage and total (grazed plus silage) for the four

time periods. The date of first and last grazing events were also included as variables. The recorded pasture average farm cover on approximately 1 December, 1 January, and 1 February according to PBI was included to denote closing average farm cover and opening average farm cover and allowed the calculation of over winter pasture growth.

*2.6. Soil Management Factors*

Soil fertility data were accounted for by including annual soil fertility test results for soil pH, soil P and K status for each paddock each year. Total N use was calculated by summing N spread according to PBI records for the four time periods; total P and total K spread were calculated similarly. The amount of N applied was approximately 250 kg N/ha per year equally across treatments, as this is the upper legal limit allowed to be spread on swards in Ireland. The amount of N fertilizer spread in each rotation was in accordance with the best management practice guidelines [28].

*2.7. White Clover Variables*

Sward WC contribution (WCc) and PRG contribution were calculated as a percentage of the total DM biomass produced at each grazing event and the average WCc for each paddock was calculated for each time period. White clover contribution to the DM offered for grazing was the main parameter used in the analysis to determine if associations existed. Paddocks were allocated to four classes of WCc based on their mean annual WC content; WCc45 $\geq$35%; 25% $\leq$ WCc30 < 35%; 15% $\leq$ WCc20 < 25%; WCc10 < 15%. White clover persistency was estimated as the slope of change in WCc over the 4 years of the experiment, calculated as a linear relationship, and placing each paddock into a separate, one of three 'Persistency (slope) Classes (PC)', dWC-4 > $-5.0$; dWC-7 $-5.0$ to $-10.0$; dWC-13 < $-10.0$, in order of declining persistence.

*2.8. Statistical Analysis*

The dataset was generated in a worksheet using the data available from the four-year experiment of McClearn et al. [18]. All variables were organised by paddock and year (40 paddocks by 4 years = 160 data points/variable). All data were analysed using SAS 9.4 software (SAS Institute Inc., Cary, NC, USA).

The first statistical approach was to investigate the relationship between WCc and factors describing management strategies, soil characteristics and meteorological conditions through multiple comparison methods such as principal component analysis and multiple regression analysis. No significant relationships were found, possibly due to the fact that the analysis was restricted to one location/site, and so the results of this analysis are not presented. Consequently, a different approach was developed, based on the WCc class proportions and slope. Data were normally distributed, linear, independent and random. Only continuous data were included as part of the analysis. Therefore, separate generalised linear models were used for each individual variable such as; PrGSH and PoGSH, PrGPM, pasture removed, soil P, nitrogen spread, winter rotation length, rainfall, SMD, temperature, radiation, pasture cover 1st of December, January and February, and sward density, with individual paddock as the experimental unit and analysed using PROC GLM.

**3. Results**

On average across all paddocks and over the 4 years, annual WCc was 24.7% with a significant and progressive decline ($p < 0.001$) from 40.6% to 25.1% to 18.3% to 14.8% in 2014, 2015, 2016 and 2017, respectively. The slope of the decline was $-3.57$, $-7.49$ and $-13.46$ for persistency class dWC-4, dWC-7 and dWC-13, respectively. Persistency class dWC-4 paddocks underwent a gentle decline compared with dWC-13 paddocks. It was also notable that the greater the initial sward WCc in the first year, the greater the slope of the decline. As a result, dWC-13 paddocks had the greatest initial WCc and slope of decline (52.0% and $-13.46$), with dWC-7 intermediate (40.0% and $-7.49$) and dWC-4 the lowest slope of decline (29.4% and $-3.57$; $p < 0.001$ for WCc and PC).

### 3.1. White Clover Contribution

The mean WCc for the four classes was 46.2%, 29.1%, 19.7% and 9.5% and are referred to as WCc45, WCc30, WCc20 and WCc10, respectively. In the early season, WCc45 and WCc30 paddocks had significantly lower pasture removed either as grazed or as total (grazed plus silage) than WCc20 and WCc10 paddocks (Table 2). White clover contribution class 45 paddocks had a greater total pasture removed in mid-season compared with WCc30, WCc20 and WCc10 paddocks. Grazed pasture removed in the late season was lower in WCc45, WCc30 and WCc10 paddocks compared with WCc20. Average pasture cover on the 1st of December, January and February was consistently lowest for WCc45 paddocks and highest for WCc10 paddocks. White clover contribution class 45 and WCc30 paddocks also had greater soil P levels than WCc20 and WCc10 paddocks. Winter rotation length was shorter for WCc45. A lower post-grazing height was recorded on WCc45 paddocks compared with WCc20 and WCc10 paddocks, with WCc30 paddocks intermediate.

**Table 2.** Comparison of paddocks in different white clover contribution (WCc) classes for pasture growth, soil phosphorus and grazing variables (only variables with significant differences ($p < 0.05$) shown).

| White Clover Contribution Class | WCc45 [1] | WCc30 | WCc20 | WCc10 | SEM | *p*-Value |
|---|---|---|---|---|---|---|
| Grazed [2] pasture removed early [3] (kg DM/ha) | 1880 [a7] | 2049 [a] | 2495 [b] | 2868 [b] | 202.5 | ** [4] |
| Grazed pasture removed late (kg DM/ha) | 2595 [a] | 2919 [a] | 3157 [b] | 2631 [a] | 142.9 | * |
| Total [5] pasture yield removed early (kg DM/ha) | 1880 [a] | 2095 [a] | 2573 [b] | 3103 [b] | 190.7 | *** |
| Total pasture yield removed mid (kg DM/ha) | 10,986 [a] | 9647 [b] | 9818 [b] | 9890 [b] | 358.3 | * |
| Pasture cover 1 December (kg DM/ha) | 547 [a] | 618 [a] | 723 [a] | 921 [b] | 69.2 | * |
| Pasture cover 1 January (kg DM/ha) | 651 [a] | 645 [a] | 849 [b] | 1091 [c] | 63.1 | *** |
| Pasture cover 1 February (kg DM/ha) | 644 [a] | 698 [a] | 897 [b] | 1171 [c] | 64.3 | *** |
| Soil phosphorus (mg/L) | 9.29 [a] | 8.75 [a] | 7.13 [b] | 6.63 [b] | 0.704 | * |
| Winter [6] rotation length (days) | 139 [a] | 148 [ab] | 153 [b] | 155 [b] | 3.6 | * |
| Post-grazing height (cm) | 3.73 [a] | 3.82 [ab] | 3.87 [b] | 3.94 [b] | 0.046 | * |

[1] WCc45 $\geq$ 35%, WCc30 $\geq$ 25 < 35%, WCc20 $\geq$ 15 < 25%, WCc10 $\leq$ 15% mean sward WC contribution. [2] Grazed = pasture yield removed as grazing. [3] early = 1 January–31 May, mid = 1 June–31 August, late = 1 September–31 December. [4] $p > 0.05$; * = $p < 0.05$; ** = $p < 0.01$; *** = $p < 0.001$. [5] total = pasture yield removed as grazing + silage. [6] Winter defined as 1 November–31 January. [7] Means within row with different superscripts are significantly different ($p < 0.05$).

### 3.2. Perennial Ryegrass Cultivars

Figure 1 illustrates that individual PRG cultivars differed in the WCc they maintained on average during the first year of the experiment and how this changed over the course of the four years ($p < 0.001$), within the overall progressive decline in WCc. Overall, the 4 year mean annual WCc for each PRG cultivar was: Glenveagh 36.4% [a]; Tyrella 28.5% [ab]; Kintyre 28.4% [ab]; Astonenergy 25.1% [b]; Dunluce, 25.6% [b]; Twymax 23.6% [b]; AberChoice 22.3% [b] and Drumbo 18.3% [bc]. The slope of decline over the four years for the individual cultivars was Tyrella −11.28 [ab], Glenveagh −9.84 [bc], AberChoice −8.46 [c], Twymax −8.32 [c], Astonenergy −8.10 [c], Kintyre −6.64 [c], Dunluce −6.44 [c] and Drumbo −6.16 [c], ($p < 0.001$, means with different superscripts are significantly different ($p < 0.05$)). Therefore, the relatively higher the initial WCc shown on the graph, the steeper the slope of the decline (e.g., Tyrella and Glenveagh compared with cultivars with a lower initial WCc such as Drumbo). Glenveagh also was the lowest yielding cultivar and was in turn the least competitive PRG cultivar. Even so, the PRG cultivars that initially had a higher WCc and declined steeply still had a higher WCc by 2017 compared with the cultivars that had lower initial WCc and lower slopes of decline. Within the overall downward trend, there was some re-ranking, particularly in 2016. This could be because 2016 was a poor year for total pasture production with the lowest annual soil temperatures of the four years.

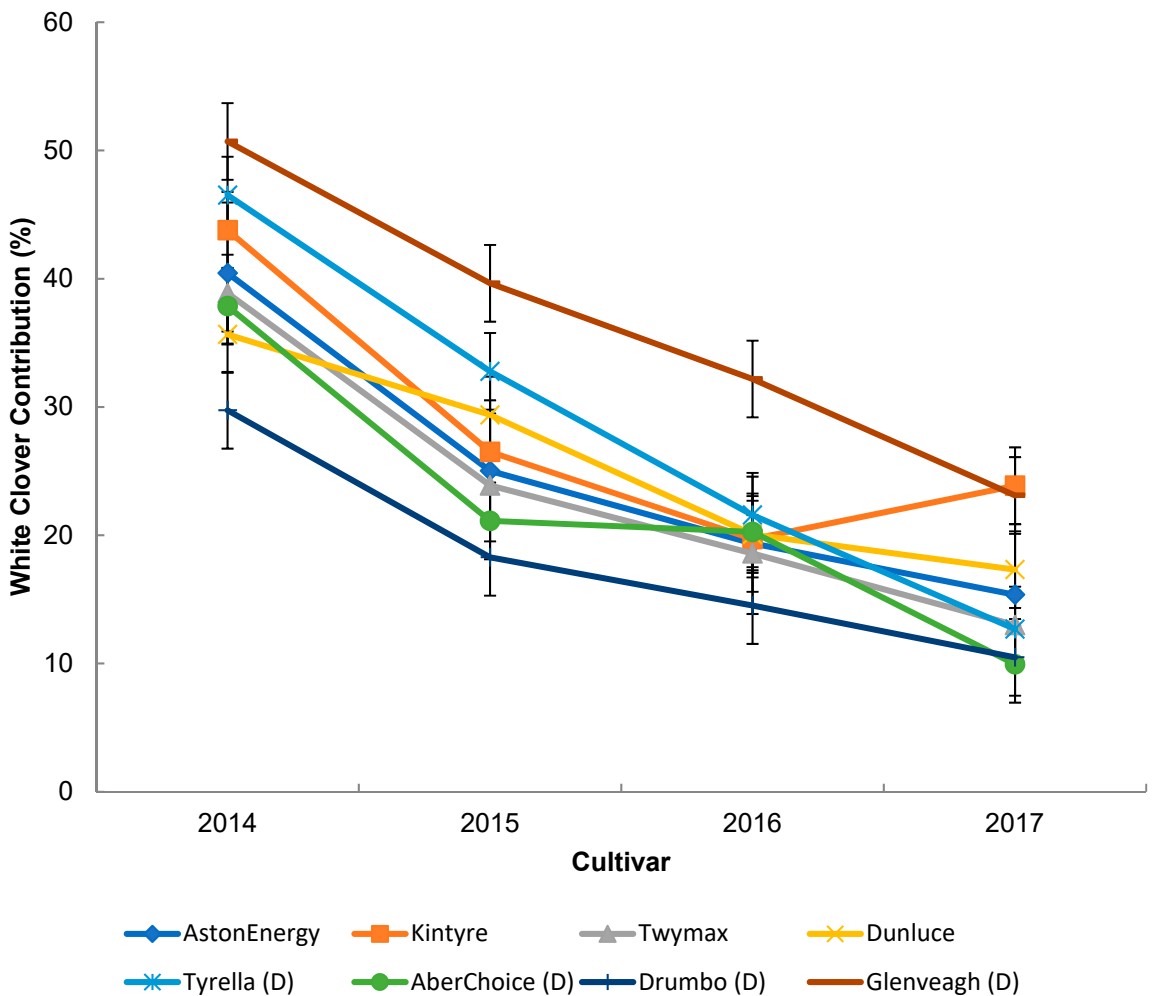

**Figure 1.** Annual white clover contribution of individual PRG cultivars.

The grazing management variables of note for the diploid and tetraploid PRG cultivars are presented in Table 3. There were no differences between the tetraploid cultivars for any of the variables with the exception of total pasture yield removed for AstonEnergy, as it had a higher pasture yield removed than Kintyre, Twymax and Dunluce. Pre- and post-grazing height, PrGPM and pasture removed were significantly lower for Tyrella and Glenveagh than AberChoice and Drumbo. Total pasture removed was significantly lower for Tyrella and Glenveagh than for Drumbo with AberChoice intermediate and not distinct. There were no significant differences in density between all the cultivars.

**Table 3.** Grazing characteristics of diploid perennial ryegrass and the mean of the tetraploid perennial ryegrass cultivars used over 4 years.

| Cultivar | Tyrella | AberChoice | Drumbo | Glenveagh | AstonEnergy | Kintyre | Twymax | Dunluce | SEM | *p*-Value |
|---|---|---|---|---|---|---|---|---|---|---|
| White clover contribution (%) | 28.4 [ac3] | 22.3 [ab] | 18.3 [b] | 36.4 [c] | 25.0 [a] | 28.4 [a] | 23.6 [a] | 25.6 [a] | ±2.99 | *** [1] |
| Pre-grazing height (cm) | 7.99 [a] | 8.82 [bc] | 9.01 [c] | 8.07 [a] | 8.32 [b] | 8.36 [b] | 8.36 [b] | 8.64 [b] | ±0.181 | *** |
| Post-grazing height (cm) | 3.79 [a] | 3.99 [a] | 4.05 [b] | 3.80 [a] | 3.72 [a] | 3.65 [a] | 3.78 [a] | 3.80 [a] | ±0.023 | * |
| Pre-grazing mass (kg DM/ha) | 1378 [a] | 1653 [bc] | 1754 [b] | 1754 [b] | 1398 [a] | 1475 [a] | 1430 [a] | 1510 [a] | ±53.6 | *** |
| Pasture yield removed (kg DM/ha) | 1434 [a] | 1637 [b] | 1736 [c] | 1438 [a] | 1509 [ab] | 1586 [ab] | 1505 [ab] | 1546 [ab] | ±51.5 | *** |
| Total [2] pasture yield removed (kg DM/ha) | 14,453 [a] | 15,894 [ab] | 17,240 [b] | 14,239 [a] | 16,153 [b] | 14,337 [a] | 15,295 [a] | 14,991 [a] | ±529.1 | *** |
| Density (kg DM/cm/ha) | 312.6 | 321.4 | 337.1 | 323.6 | 313.0 | 332.3 | 314.7 | 330.4 | ±7.66 | NS |

[1] NS = Non significant: $p > 0.05$; * = $p < 0.05$; *** = $p < 0.001$. [2] Total = pasture yield removed as grazing + silage. [3] Means within row with different superscripts are significantly different ($p < 0.05$).

### 3.3. Meteorological Data

As grazing cycles were driven by pasture availability, timing of grazing differed between paddocks in each WCc class and so the meteorological conditions experienced

before, during and after each grazing event also differed (Table 4, only variables with significant responses shown). Paddocks in WCc45 received less rainfall during the regrowth phase compared with WCc30, WCc20 and WCc10 paddocks for the full year and in the mid-season period. Paddocks in class WCc45 received less rainfall at the time of grazing in the early season compared with WCc30 and WCc10 paddocks, with no difference between WCc45 and WCc20 paddocks observed. White clover contribution class 45 paddocks also had significantly lower rainfall during the regrowth stage in both the early and mid-season compared with the other 3 classes. Rainfall during the seven-day period post-grazing for the full year was lower for WCc45 paddocks compared with WCc30 and WCc20 paddocks but similar to WCc10 paddocks. Rainfall during the seven-day period post-grazing in the mid-season was lower for WCc45 paddocks compared with all other classes. White clover contribution class 45 paddocks experienced a greater SMD than the other classes, both across the full year and in mid-season. Late season SMD was greater, and not different for WCc45, WCc30 and WCc20 paddocks, compared with WCc10 paddocks, which had the lowest SMD. Similarly, SMD was greater for WCc45 paddocks compared with the other classes for both the seven-day period pre- and post-grazing for the full year and in mid- and late seasons. White clover contribution class 45 paddocks had the highest air temperatures in the seven-day period post-grazing for the full year and in mid-season compared to the other 3 classes. Increased radiation in the seven-day period post-grazing mid-season was associated with the higher WCc in WCc45 and with the higher WCc in WCc30 compared with WCc20 and WCc10 paddocks.

**Table 4.** Comparison of incident meteorological conditions for paddocks of different white clover contributions (WCc) before, during and after each grazing from 2014–2017, summarised for different grazing periods.

| White Clover Contribution Class | WCc45 [1] | WCc30 | WCc20 | WCc10 | *p*-Value |
|---|---|---|---|---|---|
| Full Year Period 1 January–31 December | | | | | |
| White clover contribution (%) | 46.2 [a][6] ± 0.923 | 29.1 [b] ± 0.825 | 19.7 [c] ± 0.844 | 9.5 [d] ± 0.923 | *** [2] |
| Rainfall during regrowth [3] (mm) | 880 [a] ± 40.3 | 1131 [b] ± 36.0 | 1169 [b] ± 36.8 | 1166 [b] ± 40.3 | *** |
| Mean SMD [4] during grazing (mm) | 25.4 [a] ± 1.39 | 17.1 [b] ± 1.24 | 15.2 [b] ± 1.27 | 13.5 [b] ± 1.39 | *** |
| Mean temperature during grazing (°C) | 12.1 ± 0.10 | 11.9 ± 0.09 | 11.9 ± 0.10 | 11.9 ± 0.10 | NS |
| Rainfall seven-day pre-grazing (mm) | 175 ± 10.7 | 185 ± 9.6 | 173 ± 9.8 | 192 ± 10.7 | NS |
| Mean SMD seven-day pre-grazing (mm) | 17.7 [a] ± 1.32 | 9.3 [b] ± 1.18 | 13.1 [ab] ± 1.21 | 8.8 [b] ± 1.32 | * |
| Rainfall seven-day post-grazing (mm) | 146 [a] ± 10.1 | 173 [ab] ± 9.1 | 198 [b] ± 9.3 | 165 [ab] ± 10.1 | ** |
| Mean SMD seven-day post-grazing (mm) | 25.8 [a] ± 1.29 | 17.5 [b] ± 1.15 | 15.1 [b] ± 1.18 | 13.9 [bc] ± 1.29 | *** |
| Mean temperature seven-day post-grazing (°C) | 12.4 [a] ± 0.10 | 12.0 [b] ± 0.09 | 12.0 [b] ± 0.09 | 12.1 [b] ± 0.10 | * |
| Mean radiation seven-day post grazing (J) | 1512 ± 15.4 | 1512 ± 14.0 | 1477 ± 14.4 | 1469 ± 15.4 | NS |
| Early-season [5] Period 1 January–31 May | | | | | |
| Rainfall during regrowth (mm) | 423 [a] ± 37.0 | 614 [b] ± 33.1 | 620 [b] ± 33.9 | 624 [b] ± 37.0 | *** |
| Rainfall during grazing (mm) | 17.7 [a] ± 2.14 | 9.3 [b] ± 1.91 | 13.1 [ab] ± 1.96 | 8.8 [b] ± 2.14 | * |
| Mid-season Period 1 June–31 August | | | | | |
| Rainfall during regrowth (mm) | 272 [a] ± 14.2 | 318 [b] ± 12.7 | 326 [b] ± 13.0 | 324 [b] ± 14.2 | * |
| Rainfall seven-day post-grazing (mm) | 68 [a] ± 7.0 | 96 [b] ± 6.2 | 109 [b] ± 6.4 | 97 [b] ± 7.0 | *** |
| Mean SMD during grazing (mm) | 37.0 [a] ± 2.06 | 24.2 [b] ± 1.84 | 20.8 [b] ± 1.88 | 19.7 [b] ± 2.06 | *** |
| Mean SMD seven-day pre-grazing (mm) | 37.8 [a] ± 1.95 | 23.9 [b] ± 1.74 | 20.7 [b] ± 1.78 | 19.3 [b] ± 1.95 | *** |
| Mean SMD seven-day post-grazing (mm) | 37.4 [a] ± 1.93 | 24.5 [b] ± 1.72 | 20.7 [b] ± 1.76 | 19.6 [b] ± 1.93 | *** |
| Mean temperature seven-day post-grazing (°C) | 14.1 [a] ± 0.10 | 13.7 [b] ± 0.09 | 13.7 [b] ± 0.09 | 13.7 [b] ± 0.10 | ** |
| Mean radiation seven-day post-grazing (J) | 1849 [a] ± 17.4 | 1849 [a] ± 15.6 | 1797 [b] ± 15.9 | 1788 [b] ± 17.4 | ** |
| Late-season Period 1 September–31 December | | | | | |
| Mean SMD during grazing (mm) | 9.3 [a] ± 1.52 | 6.9 [a] ± 1.36 | 6.3 [a] ± 1.39 | 1.6 [b] ± 1.52 | ** |
| Mean SMD seven-day pre-grazing (mm) | 13.2 [a] ± 1.35 | 7.3 [b] ± 1.21 | 5.5 [b] ± 1.23 | 1.3 [c] ± 1.35 | *** |
| Mean SMD seven-day post-grazing (mm) | 6.6 [a] ± 1.35 | 6.9 [a] ± 1.21 | 4.9 [ab] ± 1.24 | 2.1 [b] ± 1.35 | * |

[1] WCc45 ≥ 35%, WCc30 ≥ 25 < 35%, WCc20 ≥ 15 < 25%, WCc10 ≤ 15% mean sward WCc. [2] NS = Non significant: $p > 0.05$; * = $p < 0.05$; ** = $p < 0.01$; *** = $p < 0.001$. [3] regrowth = time from cows leave paddock or silage cut to cows re-entering in the next rotation; grazing = time when cows are grazing the paddock; pre = seven days prior to cows enter paddock; post = seven days after cows finish grazing. [4] SMD = soil moisture deficit. [5] early = 1 January–31 May, mid = 1 June–31 August, late = 1 September–31 December. [6] Means within row with different superscripts are significantly different ($p < 0.05$).

### 3.4. Declining Annual Slope

The mean declining annual slope for the three classes was $-3.57$, $-7.49$ and $-13.46$ and are referred to as dWC-4, dWC-7 and dWC-13, respectively. Declining white clover-7 and dWC-13 paddocks had lower soil P levels than dWC-4 paddocks ($p < 0.01$), indicating an underlying association with declining WC persistency (Table 5, only variables with significant responses shown). The number of silage cuts was significantly higher for dWC-13 paddocks than dWC-7 paddocks, but dWC-4 paddocks were not different to either of the other two classes. The length of time before silage cuts were taken was lower for dWC-4 and dWC-7 paddocks than dWC-13. Pre-grazing height, PrGPM and pasture removed all showed a similar trend in terms of the slope of the decline of WCc. Pre-grazing height was higher for dWC-4 than dWC-13 paddocks with dWC-7 paddocks not different from dWC-4 or dWC-13 paddocks. Both the PrGPM and the pasture removed were higher for dWC-4 and dWC-7 compared with dWC-13 paddocks. Total N applied to the paddocks in the early season was lower for dWC-4 and dWC-7 than dWC-13. Total P spread was lower for dWC-4 than dWC-7 and dWC-13 paddocks, as paddocks that had lower P fertilizer spread were the paddocks that had higher soil P levels. Total grazing days/ha/yr was lowest for dWC-13 compared with dWC-4 and dWC-7 paddocks.

**Table 5.** Comparison of paddocks in different declining white clover (dWC) classes for pasture growth, soil phosphorus and grazing variables.

| White Clover Contribution Slope Class | dWC-4 [1] | dWC-7 | dWC-13 | *p*-Value |
|---|---|---|---|---|
| Soil phosphorus (mg/L) | 9.64 [a][5] ± 0.610 | 7.09 [b] ± 0.588 | 7.21 [b] ± 0.610 | ** [2] |
| Grazed [3] pasture yield removed (kg DM/ha) | 10,683 [a] ± 425.4 | 11,308 [a] ± 409.9 | 9451 [b] ± 425.4 | ** |
| Grazed pasture yield removed early [4] (kg DM/ha) | 2546 [a] ± 180.2 | 2452 [a] ± 173.6 | 1936 [b] ± 180.2 | * |
| Grazed pasture yield removed mid (kg DM/ha) | 5196 [ab] ± 320.9 | 6014 [a] ± 309.3 | 4761 [b] ± 320.9 | * |
| No. silage cuts | 1.42 [ab] ± 0.112 | 1.14 [a] ± 0.108 | 1.58 [b] ± 0.112 | * |
| Silage rotation length | 37.9 [a] ± 3.35 | 37.4 [a] ± 3.43 | 48.9 [b] ± 3.20 | * |
| Pre-grazing height (cm) | 8.70 [a] ± 0.116 | 8.45 [ab] ± 0.112 | 8.20 [b] ± 0.116 | * |
| Pre-grazing mass (kg DM/ha) | 1551 [a] ± 35.7 | 1551 [a] ± 34.4 | 1403 [b] ± 35.7 | ** |
| Pasture removed (kg DM/ha) | 1593 [a] ± 32.8 | 1599 [a] ± 31.6 | 1452 [b] ± 32.8 | ** |
| Total nitrogen early (kg/ha) | 122 [a] ± 4.3 | 118 [a] ± 4.2 | 137 [b] ± 4.3 | ** |
| Total phosphorus fertiliser spread (kg/ha) | 6.8 [a] ± 1.43 | 11.3 [b] ± 1.38 | 14.0 [b] ± 1.43 | ** |
| Total grazing days/ha/yr | 697 [a] ± 23.7 | 708 [a] ± 22.8 | 609 [b] ± 23.7 | ** |

[1] dWC-4 = $\geq$ -5, dWC-7 $\leq$ $-5$ $\geq$ $-10$, dWC-13 $< -10$, slope of annual change in clover contribution class. [2] $p > 0.05$; * = $p < 0.05$; ** = $p < 0.01$. [3] Grazed = pasture yield removed as grazing. [4] early = 1 January–31 May, mid = 1 June–31 August, late = 1 September–31 December. [5] Means within row with different superscripts are significantly different ($p < 0.05$).

Those meteorological variables that were significantly associated with the decline in annual WCc are presented in Table 6. Soil moisture deficit in the early season was greater in dWC-4 and dWC-7 paddocks than in dWC-13 paddocks. Similarly, SMD for the seven-day period pre-grazing in the early season was again highest in dWC-4 and dWC-7 paddocks and similar in dWC-13. Rainfall for the seven-day period post-grazing in mid-season was significantly less for dWC-4 but similar for dWC-7 and dWC-13 paddocks. Rainfall at the time of grazing during mid-season was lowest for dWC-4 and dWC-7 paddocks and significantly higher for dWC-13 paddocks. Radiation in the seven-day period post-grazing in the mid-season was similar for dWC-4 and dWC-7 and lowest for dWC-13.

**Table 6.** Comparison of paddocks in three different declining white clover (dWC) classes in terms of key meteorological parameters (only variables with significant differences ($p < 0.05$) shown).

| White Clover Contribution Slope Class | dWC-4 [1] | dWC-7 | dWC-13 | *p*-Value |
|---|---|---|---|---|
| Mean white clover persistency slope | −3.57 [a6] ± 0.263 | −7.49 [b] ± 0.253 | −13.46 [c] ± 0.263 | *** [2] |
| Rainfall seven-day post [3]-mid [4] (mm) | 81 [a] ± 6.0 | 98 [b] ± 5.8 | 101 [b] ± 6.0 | * |
| Rainfall grazing-mid (mm) | 22.4 [a] ± 3.25 | 27.8 [ab] ± 3.13 | 34.4 [b] ± 3.25 | * |
| Mean radiation seven-day post-mid (J) | 1840 [a] ± 14.7 | 1840 [a] ± 14.2 | 1788 [b] ± 14.7 | * |
| Mean SMD [5]-early (mm) | 7.3 [a] ± 0.87 | 7.9 [a] ± 0.83 | 3.9 [b] ± 0.87 | ** |
| Mean SMD seven-day pre-early (mm) | 4.9 [a] ± 0.75 | 5.2 [a] ± 0.72 | 2.6 [b] ± 0.75 | * |

[1] dWC-4 = ≥ −5, dWC-7 ≤ −5 ≥ −10, dWC-13 < −10, slope of annual change in clover contribution class. [2] $p > 0.05$; * = $p < 0.05$; ** = $p < 0.01$; *** = $p < 0.001$. [3] post = after cows finish grazing paddock. [4] early = 1 January–31 May, mid = 1 June–31 August, late = 1 September–31 December. [5] SMD = soil moisture deficit. [6] Means within row with different superscripts are significantly different ($p < 0.05$).

## 4. Discussion

In order to capitalise on the benefits of including WC in a PRG sward, WC must be maintained within the sward above a minimum threshold. Andrews et al. [19] hypothesised that a sward WC proportion of 20% or greater is required in order to observe an animal production response. However, the contribution of WC to the yield of a mixed sward is variable due to the competitive characteristics of WC in the sward and the effect of the continually changing environment within which WC operates [24]. In the present grazing study, there was a decrease in WCc over the four years, from a relatively high initial WCc. It is important to restate that this was a whole farm system experiment, with swards rotationally grazed by dairy cows over 4 years, and with excess grass ensiled. This contrasts with the majority of previous research that has been undertaken in plot studies [24] or under shorter term cattle or sheep grazing [31,32]. Therefore, the results of the current study are directly applicable to farm-scale rotationally grazed, intensively managed swards and identify what is within or beyond the control of the farmer. However, it is important to also note that this analysis was undertaken at only one site with relatively limited variations in meteorological and soil conditions.

### 4.1. Meteorological Factors Associated with WC Content and Persistency

Although grazing during periods of inclement weather can be avoided, it is beneficial to maximize the number of grazing days during the year. For example, Hanrahan et al. [33] reported that extra days at grass in spring and in autumn can usefully increase the profitability of grazing systems. Reduced radiation, higher rainfall post-grazing in mid-season and lower SMD in the early season were characteristics of paddocks that had a greater slope of decline in WCc. Similarly, higher rainfall during the regrowth period and in the seven days post-grazing during mid-season and generally lower SMD overall, were characteristics of paddocks that had lower annual WCc. This was not unexpected, as it is widely acknowledged that WC favours warm, dry, and bright growing conditions [34,35]. The current results, therefore, show that when deficiencies in these key meteorological factors occur, this impacts significantly on the ability of WC to persist. This is in agreement with Wachendorf et al. [24], who found that meteorological factors across 12 experimental sites set a threshold on the success of maintaining a certain level of WC in the sward. Furthermore, the existence of these interactions between meteorological factors and WC prevalence and performance suggests that they will equally modify the outcomes of management strategies. A further implication of the observed large changes in meteorological conditions that occurred from year to year at this single site is the resultant fluctuations in WC persistency and this influence is well documented by previous research [24,32,34–36]. Rainfall, SMD and temperature were the primary meteorological determinants of the WC content and persistency responses in all periods. That radiation influenced WC dynamics during mid-season was contrary to Wachendorf et al. [24] and Nolan et al. [32], who found that both winter and spring radiation were more influential than summer radiation.

This disparity was possibly because either radiation was no longer a limiting factor in these other studies, or because it had become confounded with the effect of increasing temperature during summer. This was not an impediment in the current study, particularly as temperature requirements for WC are higher than those for PRG [35]. Furthermore, Wachendorf et al. [24] have reported that temperature, radiation and rainfall were all intimately involved in determining sward WC content throughout the annual growing cycle in a largely similar way to the current study.

### 4.2. Characteristics of Paddocks with High WCc and Persistency

White clover persistency is dependent on the integrity of the plant's stolon network [37]. Hence, it is widely accepted that WC requires light to reach the base of the sward for its growing point/stolons to function at maximum capacity [38]. This maximises the plant's capacity to establish and proliferate. Therefore, the findings in the current study, that shorter rotation lengths and greater grazing days were associated with paddocks with high WCc and persistency, is consistent with the underlying biology. This further emphasises the importance of balancing grazing management practices to provide favourable conditions for WC to proliferate, as well as managing herd grazing demand. Grazing management practices, therefore, need to strike a balance between grazing being sufficiently intense for light to reach stolons to stimulate branching, and grazing intervals being sufficiently long for leaves of WC to claim their place in the upper canopy of the sward, to optimise photosynthate production and ensure sufficient energy to persist [38].

Multiple previous studies have highlighted the issues of the long-term sustainable persistency of WC at high N fertilizer levels [10,14,17]. In this study, the effect of the amount of N spread in early and mid-season was somewhat inconsistent but generally showed that greater levels of N at these times led to reduced WCc (data not shown). Greater levels of N in the early season were also characteristic of paddocks that had a greater decline in WCc. Total N application did not appear to have a substantial effect however, as a similar target of 250 kg N/ha was in place for all paddocks. A lower PoGSH was a characteristic of paddocks with a greater WCc, even though the differences in PoGSH were relatively small. This further confirms recent studies showing that by precise grazing management to control grass covers, it is possible to maintain good sward WC content, even with relatively high levels of applied N (200 kg N/ha; [8,11]).

The effect of the number of silage cuts on the rate of decline in WCc was inconsistent, but a greater decline was observed for paddocks with the highest intial WCc, and this was accentuated the longer a paddock was closed for silage. Barthram and Grant [39] suggested that climatic conditions during the conservation period could alter the effects of the large conservation stands on species composition by affecting the growth of the PRG and WC differently, and through the subsequent effects of shading. Therefore, this is likely to be an unavoidable consequence under farm-scale conditions as periods of excess sward growth that exceed herd demand will always need to be managed, as was evident in the current study.

Autumn pasture management resulting in a high pasture cover over the winter period will reduce radiation reaching the base of the sward and has been consistently proven to reduce sward WC content [38]. In the current study, there were three important pasture covers recorded at the start, in the middle and at the end of the closed period; approximately the 1 December, 1 January and 1 February. It was evident that paddocks with a greater average pasture cover recorded on those dates and/or were closed for longer over the winter period retained a lower WCc. This corroborates previous research that has shown similar findings [14,40] and confirms the importance of closing PRG-WC swards at a relatively low average pasture cover and resuming grazing in spring as soon as weather permits. Closing paddocks at low covers with a high WC content later in the final rotation in autumn will reduce the pasture cover carried over the winter and aid in the retention of WC content but requires strategic pasture management. This indicates a need for further research into the dynamics of PRG-WC mixtures to identify the optimum balance

in terms of closing strategy to support WC persistency whilst also ensuring sufficient pasture is available for the start of the grazing period in spring. Therefore, the key research challenge will be to underpin the methods of achieving relatively high levels of over-winter pasture growth without compromising the balance of companion PRG and WC content in the sward.

It is well-known that the basic building blocks of soil fertility need to be at optimum levels for PRG-WC swards to thrive [21]. Therefore, in this study, it was not surprising to see that paddocks with higher soil P status were associated with a higher WCc and a lower decline in WCc even though dWC-7 and dWC-13 paddocks received greater P fertiliser applications than dWC-4 paddocks. It is further notable that those paddocks with higher WCc were at index 4 (>8.0 mg/L) compared to index 3 (5.1–8.0 mg/L) for paddocks with significantly lower WCc, which is still considered a high soil P index. Therefore, this may be indicating a greater WC sensitivity under intensively grazed, high N managed swards, than is evident under more controlled plot studies. Further study of this observation could be worthwhile and also clearly emphasises the importance of annually testing and maintaining soil P levels for intensively managed grazing systems.

### 4.3. Grass Cultivar Responses

A notable observation was the apparent greater compatibility of the WC with some PRG companion cultivars. B. McClearn et al. [18] reported that diploid and tetraploid WC swards had similar sward WC content on average during the 4 years of that parallel study, though large variations in sward WC content amongst paddocks, seasons and over time were evident. Elgersma and Schlepers [41] also found no difference in sward WC content between tetraploid and diploid PRG cultivars and Tozer et al. [42] reported similar sward WC contents with late season tetraploid and diploid swards but lower WC content with mid-season diploid swards. In contrast, Swift et al. [43] found tetraploids to be more favourable to WC growth than diploids. Therefore, the complex interactions between PRG cultivar, WC cultivar, management and climate can lead to a variety of responses in terms of the sward WC content achieved. This complex pattern was observed in the current study as there was no difference in WC content between the individual tetraploid cultivars over the four years but there was a large difference between two of the diploid cultivars. Glenveagh consistently had a lower pre-grazing height, post-grazing height, PrGPM and pasture removed than Drumbo. This could potentially have allowed greater levels of radiation to reach the base of the sward or allowed the WC to attain a sufficiently high position in the canopy to intercept greater levels of irradiation. Given that, Glenveagh had the highest and Drumbo the lowest WC content in the study and given similar indications from earlier studies [32,44], these sward structure characteristics of a PRG cultivar would appear to be important in determining compatibility with WC. The height of the PRG canopy seems more influential than the density of the sward as both Glenveagh and Drumbo had similar densities (323 and 337 kg DM/cm/ha, respectively), which is a recognised characteristic of these cultivars. For example, on the Irish Pasture Profit Index (PPI), Glenveagh scores 6.9 and Drumbo 6.5 [45], where above 6.0 is a high density. However, Byrne et al. [46] reported that Glenveagh had the lowest spring and autumn pasture growth compared with the other cultivars in the present study, which may also have favoured WC growth. The combination of lower pre-grazing height, lower growth in spring and autumn and lower total annual DM production, as observed in this study, can again be interpreted to mean less shading of the WC laminae and greater sward WC survival, as others have reported [39,44]. The results observed in this study pose questions around the agressivity of both PRG and WC cultivars in terms of their growth habits and particularly how 'aggressive' PRG cultivars impact the subsequent WC content in a sward. Therefore, companion cultivar agronomy is a further important factor impacting on WC persistency.

Companion grass agronomy and white clover compatibility should not overshadow other aspects of sward performance. Notably, although Glenveagh was the most favourable cultivar in terms of achieving the highest mean WCc, this may not be entirely desirable,

as very high sward WC contents can pose challenges in terms of over-winter growth, grazing in spring and bloat [11,18]. Glenveagh has been removed from the PPI since 2018, where it ranked in last position largely due to its relatively low annual production. However, given that Astonenergy, Kintyre and AberChoice have consistently featured relatively highly on the PPI in previous years and had WCc of over 20% in the current study, good WC compatibility and high grass productivity are not mutually exclusive. Hence, a high performing cultivar with a moderately low PrGSH (<10 cm) and low PoGSH (approximately 4 cm), as ranked by [47], would appear preferable for mixing with WC.

## 5. Conclusions

The success of WC inclusion and persistency in PRG swards is dependent on a litany of factors, and conducting this large-scale grazing study has helped place these within the overall context of a high-intensity grazing regime. Good soil fertility remains fundamental to success of WC as below minimum levels greatly diminish WC growth and makes management strategies to improve sward WC persistency less effective. The examination of all the variables in this study point to a common theme that management strategies should focus on the manipulation of the degree of light reaching the base of the sward, as a key factor for increasing sward WC content. On this basis, autumn pasture management plays an important role in maintaining WC persistency and contribution, ideally requiring a low pasture cover with a relatively high WC content at closure. There should also be careful consideration taken to select companion PRG cultivars with low PrGSH and PoGSH characteristics to naturally aid persistency of WC in the sward. However, as discussed, these factors must be balanced with the need to maintain the overall productivity of the sward, which adds certain complexities that merit further study.

**Author Contributions:** Conceptualization, Á.M., L.D. and B.M. (Brian McCarthy); methodology, Á.M., L.D. and B.M. (Brian McCarthy); validation, Á.M., L.D., T.J.G. and B.M. (Brian McCarthy); formal analysis, Á.M. and L.D.; investigation, Á.M.; data curation, Á.M., B.M. (Brid McClearn), M.D. and C.G.; writing—original draft preparation, Á.M.; writing—review and editing, Á.M., L.D., T.J.G. and B.M. (Brian McCarthy); visualization, Á.M.; supervision, T.J.G. and B.M. (Brian McCarthy); project administration, T.J.G. and B.M. (Brian McCarthy); funding acquisition, B.M. (Brian McCarthy). All authors have read and agreed to the published version of the manuscript.

**Funding:** This research was funded by Dairy Research Ireland—Irish Dairy farmers levy funding and the Teagasc Walsh Scholarship Programme No. 0410.

**Institutional Review Board Statement:** Not applicable.

**Informed Consent Statement:** Not applicable.

**Data Availability Statement:** Research data are not shared.

**Acknowledgments:** The authors would like to gratefully acknowledge the work and the invaluable assistance of the farm and technical staff based at Teagasc Clonakilty and Teagasc Moorepark.

**Conflicts of Interest:** All authors declare no conflict of interest with the subject matter or materials discussed in this manuscript.

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
