# Peer review of "A Retrospective Analysis of White Clover (Trifolium repens L.) Content Fluctuation in Perennial Ryegrass (Lolium perenne L.) Swards under 4 Years of Intensive Rotational Dairy Grazing"

_agriculture, doi:10.3390/agriculture12040549_

Round 1
Reviewer 1 Report
I really enjoyed this paper and was quite informed by it. It presents a lot of data and interesting analyses examining factors affecting white clover contributions and persistency in perennial ryegrass swards under rotational stocking. Below I list some relatively minor recommendations for improving clarity.
Line 41. Make it plural “possesses” considering white clover is singular, i.e., one species.
Line 43. Consider a slight change to “while allowing a reduction” or “thereby allowing a reduction” to improve clarity of this sentence.
Line 44. Change to “is nutritionally superior forage to PRG” and use singular “intake”.
Line 74. Change to “associated with WC persistency”, i.e., delete “the” and “decline”.
Line 82. Please list “Ireland” here to indicate the country where the research took place.
Line 102. What’s in the concentrate? Please identify the composition. Does it add anything to the pasture through the animal with respect to nutrient cycling?
Line 137. Should sward density be in kg DM per cubic cm?
Line 161. Change to “data were collected”.
Line 164. Change to “are presented”.
Line 165. Consider showing the formula or explaining how SMD was computed considering its in explaining WC contribution and would inform readers beyond providing a reference.
Line 207 and 213. Consider changing “grazing’s” to “grazing events”.
Line 208. Please clarify whether a “previous event” refers to grazing, silage cut, or both.
Line 214. Delete “the” before December.
Line 220. Change to “data were”.
Line 222. Replace the comma with a semicolon.
Line 223. Change to “spread were”.
Line 230. Delete the extra spaces before PRG and change “was” to “were”.
Line 264. The abbreviation for PC hasn’t been defined. It looks like it might be first used in line 238: Persistency Class.
Table 2. Reduce spacing between “pasture” and “cover” in Jan and Feb rows.
Table 2. The Tetraploid mean in the white clover contribution row lacks a superscript.
Line 325. Reduce spacing between sentences.
Line 337. Delete “In” and start sentence with “Late season SMD”.
Line 544. Consider changing this to “Companion grass agronomy and white clover compatibility”. Use of the /clover comes across as colloquial.
Line 555. The end of the sentence needs a period.
Line 559. The sentence is unclear. Consider starting with “Good soil fertility remains fundamental to success of WC as below minimum levels greatly diminish WC growth and makes management strategies to improve sward WC persistency less effective.
L567. Check font size of the text in this line relative to surrounding lines. The bottom part of this line is cut off.
Reviewer 2 Report
This manuscript was well written. The objective of the study is well addressed. The conclusion is very concise and useful. However, the statistical method part was too short and needed to provide more details.
- Line 252. The authors stated “individual variable”. It would be helpful if the authors provided more specific of individual variable. It can be “individual variable such as …..” to make the audiences understand well about the statistical model.
- Provide detail for statistical assumption checking. The GLM requires various assumptions and the data used in this study is subjected to be tested.
- The generalized linear model is addressed for the analysis. Thus, it was assumed that not only continuous data is used, but also other types of data because of the term “generalized”. Thus, if the models were not clearly stated, some confusion may arise. Consistent with mention#1, the model should be clearly stated. For example, what are continuous variables, what is count variable?
- What kind of multiple comparisons method used? Tukey’s test? Bonferroni test?
- Table 3, Table 4 and Table 5, the author used pool SED. When these values are used, the audience will have no information of descriptive statistics. I recommended using mean +/- SE for each group. The pool SE is good, if one would like to see the pool SE used for GLM.
- The letter a,b and 4 in Table 2 should be a superscript letters.
- The presentation of results in the main text form was overlapped with the table form. The author indicated several p values in the main text (e.g. line 356-373) that the audiences can find in the Table. Thus, it is not necessary to repeat these results.
Reviewer 3 Report
This manuscript try to examine fluctuations in white clover content in perennial ryegrass swards within a high nitrogen
input grazing dairy system using a larger, overall systems experiment. However, there are many factors which make me very hard to follow how authors set replication. As I notice, eight perennial ryegrass cultivars were sown individually with two white clover cultivars which repeat 5 times and have total 40 paddocks. How other variations like pasture growth, soil phosphorus and grazing variables were compared if the classification is not even and data will be not balance. And also, how will you conduct the analysis about comparison of paddocks in different white clover contribution (WCc) classes if the classification number is higher different. Even there were no differences between the tetraploid cultivars only a tetraploid mean is presented Grazing characteristics of diploid perennial ryegrass and tetraploid perennial ryegrass cultivars should present individual rather than only use the mean of the tetraploid cultivar.
Round 2
Reviewer 3 Report
The author's reponse has address my questions and I am happy to suggest accept this manuscript.